# Low Prevalence of HLA-G Antibodies in Lung Transplant Patients Detected using MAIPA-Adapted Protocol

**DOI:** 10.3390/ijms242216479

**Published:** 2023-11-18

**Authors:** Pascal Pedini, Lucas Hubert, Federico Carlini, Jean Baptiste Baudey, Audrey Tous, Francois Jordier, Agnès Basire, Claude Bagnis, Martine Reynaud-Gaubert, Benjamin Coiffard, Jacques Chiaroni, Monique Silvy, Christophe Picard

**Affiliations:** 1Immunogenetics Laboratory, Etablissement Français du Sang, 13005 Marseille, Francechristophe.picard@efs.sante.fr (C.P.); 2ADES UMR 7268, Aix Marseille University, 13005 Marseille, France; audrey.charlier@efs.sante.fr (A.T.);; 3IRCCS Ospedale Policlinico San Martino, 16100 Genova, Italy; 4Lung Transplant Department, Aix-Marseille University, APHM, 13015 Marseille, France

**Keywords:** HLA-G, antibodies, lung transplantation

## Abstract

Lung transplantation is often complicated by acute and/or chronic rejection leading to graft-function loss. In addition to the HLA donor-specific antibodies (HLA-DSA), a few autoantibodies are correlated with the occurrence of these complications. Recently, antibodies directed against non-classical HLA molecules, HLA-G, -E, and -F have been detected in autoimmune diseases, like systemic lupus erythematosus. Non-classical HLA molecules are crucial in the immunological acceptance of the lung graft, and some of their isoforms, like HLA-G*01:04 and -G*01:06, are associated with a negative clinical outcome. The aim of this study is to determine the frequency of detection of HLA-G antibodies in lung transplant recipients (LTRs) and their impact on the occurrence of clinical complications. After incubating the cell lines SPI-801, with and without three different HLA-G isoform expression, with sera from 90 healthy blood donors and 35 LTRs (before and after transplantation), HLA-G reactivity was revealed using reagents from commercial monoclonal antibody immobilization of platelet antigen assay (MAIPA ApDIA^®^). Only one serum from one blood donor had specific reactivity against the HLA-G transduced lines. Non-specific reactivity in many sera from LTRs was observed with transduced- and wild-type cell lines, which may suggest recognition of an autoantigen expressed by the SPI-801 cell line. In conclusion, this study allowed the development of a specific detection tool for non-denatured HLA-G antibodies. These antibodies seem uncommon, both in healthy subjects and in complicated LTRs. This study should be extended to patients suffering from autoimmune diseases as well as kidney and heart transplant recipients.

## 1. Introduction

Lung transplantation (LT) is a therapeutic option for chronic, irreversible respiratory failure. Survival after LT has steadily increased over time, but the incidence of mild and long-term morbidity and lung dysfunction remains very high due to acute or chronic rejection [1]. These clinical complications may be linked to immune dysregulation, implying humoral and cellular autoimmunity and/or alloimmunity. Recent works have suggested that autoimmune reactivity is a key element in the occurrence of chronic graft rejection [2,3,4]. Antibodies directed against Tubulin K-1, a membrane protein expressed by bronchial epithelial cells, and against collagen type V, an extracellular matrix protein normally sequestered in bronchial tissue, are strongly correlated with the occurrence of bronchiolitis obliterans syndrome (BOS) after LT [5]. This production of autoantibodies may be secondary to the exposure of cryptic antigens of the self, following tissue remodeling after LT, to the expression of neo-antigens in an inflammatory context, or to the stimulation of a cross-reaction by antigenic mimicry between pathogens and self-antigens [6]. More recently, it has been shown that the release of graft exosomes in an inflammatory context increases the risk of autoimmunity [7].

The HLA-G molecule has been suggested as a prognostic biomarker of LT outcome by two independent studies [8,9]. HLA-G has both humoral and cellular anti-inflammatory immunosuppressive properties, in particular by inhibiting NK and cytotoxic T lymphocyte (CTL)-mediated activity as well as B-cell activation via their inhibitory receptor (ILT-2, -4, and KIR2DL4) [10]. This non-classical HLA class I molecule also acts indirectly on immune control as HLA-E preferentially loads its signal peptide [11,12]. HLA-G’s anti-inflammatory potential is supported by its increased expression through anti-inflammatory cytokines, as IL-10, in a positive feedback mechanism; by its increased expression in autoimmune diseases; and by its involvement in promoting viral- or parasitic-infection escape [10].

Interestingly, HLA-G is expressed at bronchial cell membranes after interferon-β stimulation [13]. Moreover, membrane-bound HLA-G expression in lung biopsies and sHLA-G expression in bronchial alveolar fluid, but not in serum, were higher in clinically stable lung transplant recipients (LTRs) than in those developing acute rejection [8,13,14].

One hundred and seventeen HLA-G alleles are currently identified, and five protein isoforms (HLA-G*01:01, *01:03, *01:04, *01:05N, 01:06) are found with a frequency >5% in all populations [15,16]. Our team, and others, have shown that these alleles are associated with differential sHLA-G levels [15,17,18].

HLA-G alleles have also been associated with clinical outcomes: the HLA-G*01:06~UTR2 haplotype has been correlated with the pejorative evolution of cystic fibrosis, and the HLA-G*01:04~UTR3 haplotype with an increase of chronic rejection, production of HLA antibodies, and a decrease in patient survival [9].

Despite its low diversity and because of its epithelial expression, HLA-G may elicit immunization mechanisms like those observed for other HLA class I molecules.

Anti-HLA-G antibodies were recently detected in patients with systemic lupus erythematosus (SLE) in a study conducted on 69 German and 29 Mexican SLE patients and 17 German healthy individuals [19,20]. Multiplex Luminex^®^-based flow cytometry was used to screen sera for antibodies directed against non-classical HLA (HLA-G, -E, and -F) and β2m. Interestingly, anti-HLA-G IgG antibodies were detected in 30% of healthy subjects (6/17) and were more frequent than those directed against HLA-E and HLA-F. In addition, anti-HLA-G antibody levels were stable in five subjects over 6 months.

Our hypothesis is that HLA-G antibodies produced following LT may interfere with the HLA-G receptors and disinhibit the cellular response directed against the lung graft. Such a mechanism would participate in a persistent inflammatory response and eventually lead to respiratory failure. Also, these anti-HLA-G antibodies can actively participate in the occurrence or progression of CLAD in transplant patients.

The main goals of this study are (1) to detect HLA-G antibodies expressed in serum using an adapted MAIPA ApDia^®^ kit and (2) to evaluate their impact on the occurrence of clinical complications in 35 LTRs.

## 2. Results

### 2.1. Population Characteristics

Ninety sera were collected from healthy blood donors (median age = 40 years (18–65); sex ratio = 0.54) with no history of transfusions, organ transplantation, or recent infections. Thirty out of forty-one women (73%) had had at least one pregnancy.

All lung transplant recipients (median age: 40 years [19–66]; sex ratio = 0.51) underwent LTx at the Marseille Lung Transplant Center. They received a first LT (8 single LTx; 26 bilateral LTx) for cystic fibrosis (49%), emphysema (20%), pulmonary fibrosis (26%), or another diagnosis (5%). One-hundred and forty-four sera were collected from 35 lung transplant recipients before and after transplant (at day 0 (D0) N = 30, at day 15 (D15) N = 25, at one month (M1) N = 28, at three months (M3) N = 32, and at twelve months (M12) N = 29). Seven LTRs had acute rejection, and seven developed chronic rejection in the first year. Seventeen produced the Donor-Specific Antibody (DSA) after LTx. One recipient experienced an acute rejection associated with DSA detection. Population characteristics are summarized in Table 1.

### 2.2. HLA-G Cell Line Transduction

The specific expression of the HLA-G isoform for each transduced cell line was assessed with Western blot, using 4H84 antibody and with flow cytometry using MEM/G9 antibody (Figure 1). Flow cytometry showed specific HLA-G membrane expression in HLA-G transduced cells, and no signal was detected for the MEM/G9 in transduced WT cells.

### 2.3. HLA-G Antibody Detection in Healthy Donors

Sera from 90 healthy donors were tested (Table 2). The screening pool ODs mean was 0.07 +/− 0.012 for donors, with no statistical difference between men and women (0.071 +/− 0.021 vs. 0.074 +/− 0.023; *p* = 0.45). The reactivity of the sera against the WT cell line was on average 0.066 +/− 0.02, corresponding to an OD ratio mean between the HLA-G pool screening and the WT cell line of 1.02. One serum was positive for HLA-G antibody detection: a 22-year-old woman with A+ blood group. A 30-year-old man, with A+ blood group, displayed a pool reactivity OD > 150 with a ratio close to one between the pool and the WT cell line (Table 2).

### 2.4. HLA-G Antibody Detection in Lung Transplantation Recipients

The pre-transplant sera (D0) and the post-transplant sera (from D15 to M12) from 35 LTRs were analyzed (Table 3). One LTR serum (ID = 9) displayed positive results from the screening test at D15 and M1. However, these results could not be replicated for the two sera in the screening test, and no positive result could be obtained in the identification assay. HLA-G antibody detection was negative for all LTR sera.

Sera from 11 LTRs was reactive with an OD > 150 for the HLA-G transduced line pool, but with a ratio close to one between the HLA-G transduced and wild cell lines, signifying the detection of antibodies against an unidentified cell antigen expressed by the SPI-6 cell line. These cases were reproducible, either in the screening assay, identification assay, or both. This non-specific reactivity was detected in seven sera before transplantation, in one serum at D15 after transplantation, in five sera at M1, and in two sera at M3. This non-specific reactivity was reproducible at different times in 3 LTRs (ID = 23 at D0 and M1; ID = 25 at M1 and M3; ID = 9 at D15 and M1). This latter patient was transplanted for emphysema and produced DSA before LT, which persisted until M3 with a Mean Fluorescence Intensity (MFI) varying from 13,000 to 7000. Patient ID = 23 was transplanted for bronchial dysplasia. No humoral event was detected until M12. Patient ID = 25 was transplanted for pulmonary fibrosis and produced DQ7 DSA with a stable MFI at 7000 from D15 to M12. None of these three patients displayed acute rejection in the first year. Patient ID = 23 reported chronic rejection. The blood groups of the three patients were different. The data are summarized in Table 3 and in Appendix A.

## 3. Discussion

Detection of HLA-G antibodies is poorly studied, although they may interfere with the immune regulation mechanisms that occur during and after organ transplant. In this study, we aimed to validate a protocol for detecting HLA-G antibodies in serum and to explore their impact on LTx outcomes. We used an adaptation assay from the MAIPA ApDIA^®^ kit to analyze 234 sera from healthy donors and lung transplantation recipients. Surprisingly, the HLA control from MAIPA ApDIA ^®^ kit was HLA-G reactive. This control is produced from the plasma of HLA hyperimmunized donors, the origin of which is unknown. Interestingly, it is not reactive for HLA-E suggesting a HLA-G-specific detection. However, a cross-reactivity with epitopes from classical HLA should not be excluded. 

Only 1 out of 234 sera was found to be reactive against all HLA-G molecules isoforms, suggesting the presence of HLA-G autoantibodies rather than alloantibodies. HLA-G antibodies were detected in one healthy donor, a 22-year-old woman (A+ blood group), without any risk factors for HLA alloimmunization, notably pregnancy. However, HLA-G antibodies were not detected in the 30 female healthy donors who had at least one pregnancy. Like classical HLA class I antibodies, HLA-G antibodies may gradually disappear only one month after delivery in first pregnancy and increase with the number of pregnancies. However, considering the crucial role of HLA-G in pregnancy success, HLA-G antibody may be more frequent in women who underwent pregnancy complications such as pre-eclampsia or unexplained fetal loss in the second and third trimester. Sera collected from pregnant women at the onset of complications and away from childbirth should be tested. The absence of detection of anti-HLA-G antibodies in patients who have not had immunizing events suggests the absence of reactivity with non-denaturated HLA antibodies. In addition, this result is surely more physiologically compatible with the risk of pathogenicity of these antibodies on the dysregulation of tolerance of the immune system than those obtained using multiplex Luminex^®^-based flow cytometry.

Thus, we could not confirm previous results, from Jucaud et al., that detected HLA-G antibodies in 6 out of 17 healthy subjects using the Luminex method [19]. Of note, our cohort size was greater. The MAIPA test may be less sensitive than the Luminex method [21]. However, classical HLA class I alloantibodies’ detection from platelet lysis using the MAIPA technique (ApDIA^®^) is as sensitive as the Luminex technique [21]. The HLA-G-antibody calibrated standard would allow sensitivity comparison. 

Luminex is a very sensitive assay for HLA antibodies’ detection, as illustrated by HLA antibodies’ identification in non-alloimmunized healthy males [22]. HLA cryptic epitopes may be exposed following antigens’ denaturation through β2m loss during the experiment process [23], as reported for HLA [24]. The main assumption concerning the detection of these HLA antibodies without an immunizing phenomenon is of cross-reactivity with epitopes of other molecules such as bacterial or animal proteins or with non-classical HLA molecules [25].

Conversely, the adapted MAIPA assay detects the complete form of HLA-G associated with β2m. However, the lack of reactivity in our adapted MAIPA assay could be due to binding competition between the MEM-G/9 antibody and those present in the sera. The MEM-G/9 monoclonal antibody was developed against the human recombinant full-length HLA-G1 protein and is today very well characterized [26]. MEM-G/9 is used in HLA-G ELISA as a capture antibody. MEM-G/9 binds cell-surface-expressed HLA-G1 monomer and dimer, HLA-G5 isoform, and also HLA-G3 isoform, suggesting that its epitope is present in the α1 domain of HLA-G [27]. So, MEM-G/9 binds to surface-expressed native HLA-G1 monomer and dimer, but not to denatured HLA-G1 [28]. 

MEM-G/9 only recognizes HLA-G1 isoforms associated with two microglobulins as shown by a mild acid treatment which has negative detection on the cell surface. However, these characterizations cannot rule out that MEM-G/9 may detect an epitope that may be masked by different isoforms. Moreover, no reactivity of MEM-G/9 was demonstrated on the HLA class I beads from ONE LAMBDA LABScreen and Immucor Lifecode^®^ (Appendix A). These data suggest that, unlike HLA-E monoclonal antibodies, 3D12 and MEM-E/02, MEM-G/9 does not cross-react with classical HLA class I molecules that are most frequent in the European population, and that its epitope may be specific to the HLA-G molecule. Furthermore, MEM-G/9 antibody did not cross-react with any of the HLA-E and HLA-F expressed on SPI-8 cell lines by FCM. However, MAIPA limits these cross-reactivities. Indeed, technically, the first deposition of serums on the transduced cells followed by washing limits the risk of cross-contamination with circulant HLA. 

No HLA-G antibody was detected in sera collected at different times before and after transplantation from 35 LTRs, strongly suggesting that anti-HLA-G antibodies are not involved in the occurrence of lung transplant complications. Our hypothesis was to detect more HLA-G autoantibodies than HLA-G alloantibodies, although DSA detection was frequent in LTRs. Indeed, the occurrence of anti-HLA-G antibodies could be secondary to the increase in expression of HLA-G by the lung graft during an inflammatory syndrome. In our LTR cohort, only three patients had HLA-G phenotyping corresponding to low HLA-G expression. The UTR2, UTR5, and UTR7 haplotypes are associated with a decrease in cellular and soluble HLA-G expression [18]. This weak expression can limit the production of anti-HLA-G antibodies directed against the graft. Similarly, only six donors had the haplotype of poor prognosis for the occurrence of chronic rejection. Furthermore, the HLA-G variability is low, and HLA-G antibodies have been detected in autoimmune diseases. However, in the latter cases, it cannot be completely excluded that this detection may rely on cross-reactivity with some epitopes expressed on HLA-E molecules or another non-classical and classical HLA. Finally, since data on the donor HLA-G status were missing, the confirmation of the alloimmunization process could also not have been considered.

Twelve LTRs (and one healthy donor) had non-specific reactivity against the non-transduced HLA-G cell lines following the MEM-G9 immunocapture. These results were accurate for three patients. One possibility is that the MEM-G9 monoclonal antibody can cross-reactivate with epitopes carried by classical HLA. However, this detection was not correlated with DSA detection, and the selected line did not express any HLA molecule. Whatever the reason for this increase in non-specific reactivity, non-transfected line control is necessary.

Finally, transplantation involves extensive injury-induced inflammation, during which process cells get activated and HLA is expressed as a monomer not associated with B2-microglobulin. These monomers expressed on cells both in tissues and in circulating exosomes expose cryptic epitopes masked by B2M when they are intact. Such exposure induces immune recognition and production of antibodies not only against allele-specific HLA monomers but also epitopes shared by all HLAs [29]. Our adapted MAIPA is not capable of detecting these types of HLA antibodies, MEM-G9 binding only to non-denatured HLA-G monomers or dimers. In contrast, multiplex Luminex^®^-based flow cytometry can detect these antibodies; the antigens fixed on the beads can be partly denatured and have a different structure. 

Notwithstanding the limits of the adapted MAIPA assay, our results supported that the production of antibodies against the various non-denatured isoforms of HLA-G is rare in non-exposed individuals (healthy donors) and in alloimmunized patients (LTRs). Such results need to be confirmed by immuno-capture performed with other HLA-G monoclonal antibodies and with other multicentric cohorts.

## 4. Materials and Methods

### 4.1. Samples

Two-hundred and thirty-four samples were analyzed from healthy donors (N = 90) and LTx patients (N = 35). LTx Patients were longitudinally followed every 6 months after LT. Inclusion criteria included at least 3 years of follow-up unless the diagnosis of CLAD was made before 3 years. Patients who died within the first 3 months post-transplantation or of a non-CLAD-related cause were excluded. 

Sera from blood donors were collected after the medical interview and before blood donation. Sera from LTRs were collected serially before transplantation and then at 15 days (D15), 1 month (M1), 3 months (M3), and 12 months (M12) after transplantation. All LTRs were genotyped for classical HLA (HLA-A, -B, -C, -DRB1, -DQB1, and -DPB1) using NGS Omixon protocol (Omixon Biocomputing Ltd., Budapest, Hungary) and for HLA-G using NG-MIX (EFS, Saint-Denis, France).

Blood donations were collected in the “Etablissement Francais du Sang”, in accordance with BSL-2 practices. A medical interview was carried out prior to blood donation to exclude donors with medical contraindications. This study was carried out in accordance with the French Public Health Code (art L1221-1), approved by the institutional ethics committee and conducted in compliance with the Good Clinical Practice Guidelines, Declaration of Helsinki and Istanbul. All lung transplant recipients in this study were from a French cohort (COLT, Cohort in Lung Transplantation, l’Institut du Thorax, INSERMUMR1087/CNRS UMR 6291, CNIL 911142) and gave their written informed consent to participate in the study in accordance with the Declaration of Helsinki. These patients were sampled between 2009 and 2014, with clinical data collected in 2017.

### 4.2. HLA-G and HLA-G K562 Cell Line Transduction and Expression Assessment

In order to cover 81% of HLA-G haplotypes [16], cDNA coding for HLA-G*01:01 (IMGT/HLA00939), -G*01:04 (IMGT/HLA00949), and -G*01:06 (IMGT/HLA01357) were obtained from Life Technologies (France) and cloned into the pWPXL lentiviral vector (Appendix A). The vector pWPXL-EGFP was carried out as a control. Lentiviral particles were generated using HEK 293T cells at 80% confluence (DSMZ, Brunswick), and were co-transfected with each lentiviral vector pWPXL, the vesicular stomatitis virus-G-encoding plasmid pMDG and the packaging plasmid pCMVΔR8.91 [30]. Lentiviral particles were collected from day 2 (D2) to 4 (D4) and concentrated in 10% polyethylene glycol. SPI-801 cells (5.104) derived from K562 cells (DSMZ ACC-86, Brunswick, Germany) were transduced using each lentiviral particle. This cell line was chosen because it does not express classical and non-classical HLA molecules and has already been the subject of several molecular transductions [31,32].

Non-infected (WT) and transduced SPI-801 cells were screened for HLA-G and HLA-E expression using Western blot and flow cytometry (Appendix A). Western blot was performed with 4H84 antibody (HLA-G, R&D system #MABF2169). Flow cytometry was performed using a Myltenyi VYB cytometer with MEM-G/9 fluorescein allophycocianin–conjugated (APC) antibody (HLA-G; Thermo Fisher #MA1-19014) or the isotype-matched control-APC Ab. HLA-G-SPI-801 cells were gated on cell populations, based upon forward and side scatter, and results were expressed as % cells expressing HLA-G above the level of fluorescence detected with the isotype control mAb (Appendix A).

### 4.3. HLA-G Antibodies Detection

HLA-G antibodies were screened using an adapted MAIPA protocol, in accordance with the MAIPA ApDIA^®^ kit. This kit allows the detection and identification of anti-platelet glycoprotein autoantibodies or anti-HPA (human platelet antigen) alloantibodies in serum using a platelet panel and platelet glycoprotein antigen immobilized using a monoclonal antibody. For the purpose of the present study, all MAIPA ApDIA^®^ kit reagents were used, except for the platelet panel and platelet glycoprotein antigen, which were replaced by HLA-G transduced cell lines and WT.

Briefly, the adapted assay (Figure 2A) consisted of two consecutive incubations: sera with HLA-G transduced cell line and then with MEM-G/9 monoclonal antibody. Cells were lysed and solubilized complexes were immobilized on a plate coated with anti-mouse antibody. Complexes were revealed using immunoassay colorimetry.

Serum HLA-G antibodies were detected with a pool of the three HLA-G transduced and WT cells and identified with each HLA-G transduced cell and WT cell. Negative control (Contr Neg), positive control (Contr HLA), both supplied with the MAIPA ApDIA^®^ kit, and two blank controls were included in each assay (Figure 2B).

Detection and identification protocol consisted of incubation (30 ± 5 min at 36 ± 1 °C) of transduced or WT cells (N = 1 × 10^6^), washed and adjusted in each well in PBS/1% BSA/0.33% EDTA with the positive control (Contr HLA) or negative control (Contr Neg), (50 µL dilution 1/1000) or with serum (50 µL dilution 1/5). Microplates were centrifuged (1000× *g*, 3 min), and each well was washed three times (ELISA Wash Buffer). MEM-G/9 monoclonal antibody was added (50 μg) and incubated for 30 ± 5 min at 36 ± 1 °C.

Cells were washed as described above and lysed (platelet lysis buffer containing triton, 130 µL, 15 min at 2–8 °C). Cell lysis supernatant (100 μL) containing the solubilized captured complex (MEM-G9/HLA-G/sera or control anti-HLA-G) were centrifuged and were incubated in a goat anti-mouse IgG-coated microplate (30 ± 5 min at 36 ± 1 °C). Plates were washed as described above, and immobilized complexes were revealed using a peroxidase reaction (substrate solution Chrom, 15 min at 36 ± 1 °C in the dark). The reaction was stopped (stop solution, 100 μL) and read with optical density (OD) using a spectrophotometer (450 nm with reference filter 650 nm).

The assay was validated when OD values were below 0.1 for the negative control (Contr Neg) and above 2 for the positive control (Contr HLA). The cut-off OD value was set at 0.15 (Contr Neg mean OD values adjusted with blank plus 3xStandard Deviation). OD values above the cut-off were considered positive. The signal-to-noise (S/N) ratio was calculated by dividing the OD value of HLA-G or HLA-E wells by that of WT cells. Serum was considered positive for HLA-G antibodies presence when the S/N ratio was >1.5.

Screening assays were performed for each serum. Identification was performed for sera with a HLA-G screening pool and OD values > 0.150, and S/N ratio > 1.5.

## Figures and Tables

**Figure 1 ijms-24-16479-f001:**
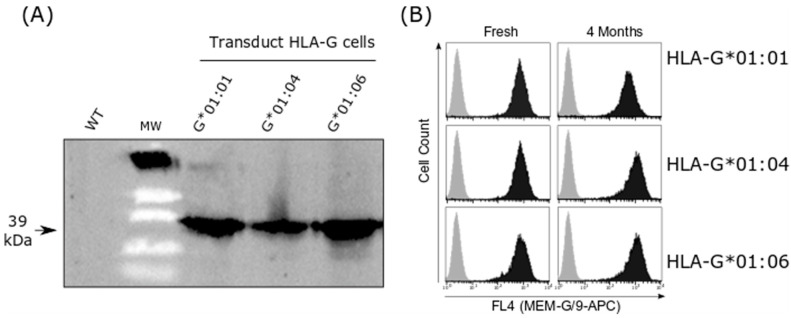
Expression of HLA-G isoform on SPI801 cell lines. (**A**) Western blot analysis of HLA-G in the transduced cells in comparison with non-transduced cells line (WT). (**B**) Flow cytometry results showing stable expression levels of HLA-G isoforms in transduced cell lines (black) and non-transduced cell lines (gray).

**Figure 2 ijms-24-16479-f002:**
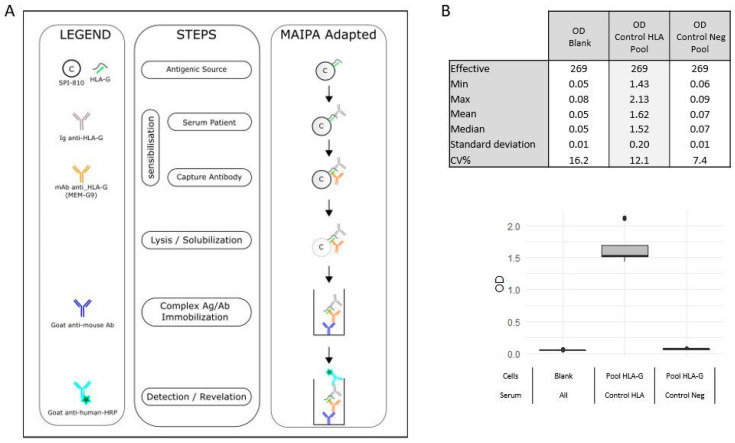
(**A**). Illustration of the different technical steps, adaptations, and compounds for MAIPA-adapted process. (**B**). Characteristics of MAIPA-adapted quality controls.

**Table 1 ijms-24-16479-t001:** Clinical and biological characteristics of all cohort LTRs, LTRs with at least one positive serum detected on transduced and non-transduced cell lines, and healthy subjects (LTR: lung transplant recipient, DSA: donor-specific antibody).

	Cohort	LTRs	Healthy
	LTRs	With Positive Serum	Blood
Patient/donor			
Number	35	12	90
Age			
mean	40	36.5	39.5
min-max	19–66	21–66	18–69
<40 years	18	7	53
≥40 years	17	5	37
Sexe			
M	17	5	41
F	18	7	49
Pregnancy	ND	ND	30
First pregancy	ND	ND	
>4 pregnancies	ND	ND	
Blood Group			
A	12	4	35
B	7	3	7
AB	2	2	8
O	14	3	40
Pathology			
Emphysema	7	3	
Cystic Fibrosis	17	5	
Fibrosis	9	3	
Bronchial dysplasia	1	1	
Histocytosis	1	0	
Type of LTx			
Single LTx	9	3	
Bilateral LTx	26	9	
Mismatch HLA			
0	5	2	
1–4	2	1	
5–8	28	9	
CMV Status			
Absence mismatch	25	8	
D+ and R− without seroconversion	1	0	
D+ and R− with seroconversion	4	1	
D− and R+	5	3	
Type of rejection			
Acute (DSA)	7	2	
Chronic (DSA)	7	1	
HLA antibodies			
DSA after LTx	26	8	
Ab No DSA after LTx	11	5	
DO	2	1	
M1	5	2	
M3	6	2	
M12	2	0	
HLA-G genotyping			
G*01:01,*--	21	6	
G*01:01,G*01:06	4	3	
G*01:01,G*01:04	6	2	
G*01:01,G*01:05N	2	0	
G*01:01,G*01:03	1	0	
G*01:03,G*01:06	1	1	

**Table 2 ijms-24-16479-t002:** Results of adapted MAIPA on healthy subjects.

	Women	Men	Post-Partum Women	Total
Effective	49	41	16	105
Serum reactivity vs. Pool HLA-G cells				
Negative	49	40	16	103
Positive	1	1	0	2
Serum reactivity (screnning OD > 0.150 and S/N ratio > 1.5) vs. identification				
Negative	0	1	0	1
Positive	1	0	0	1

**Table 3 ijms-24-16479-t003:** Results of adapted MAIPA in LTRs cohort at D0, D15, M1, M3, and M12 after LTx. Positive threshold: OD > 0.150 (OD: optical density, WT: wild type, and ND: not determined).

	Collection Dates (D: Day; M: Month)										Characteristic of Pathology							Typage HLA-G	
	OD		D15		M1		M3		M12										
Patients	OD	OD Pool	OD	OD Pool	OD	OD Pool	OD	OD Pool	OD	OD Pool	Disease	Acute Rejection	Chronic Rejection	DSA	CMV Status	Type of LTx	HLA	HLA-G	Haplotypes UTR
	WT Cells	HLA-G Cells	WT Cells	HLA-G Cells	WT Cells	HLA-G Cells	WT Cells	HLA-G Cells	WT Cells	HLA-G Cells							Mismatch	Genotyping	
1	NR	NR	0.055	0.066	0.054	0.058	0.052	0.073	0.055	0.056	Histiocytosis		Yes	D15	0	bilateral	8	G*01:01,*--	UTR2,UTR4
2	0.099	0.082	0.059	0.071	0.071	0.061	0.085	0.058	0.07	0.06	Cystic Fibrosis				0	bilateral	7	G*01:01,*--	UTR1,UTR1
3	0.062	0.058	0.051	0.063	NR	NR	0.074	0.06	0.059	0.058	Fibrosis			M1	0	bilateral	8	G*01:01,*--	UTR1,UTR1
4	0.115	0.136	*0.154*	0.127	0.115	0.106	0.06	0.063	0.084	0.092	Fibrosis	M12		D15	3	bilateral	0	G*01:01,*--	UTR1,UTR1
5	*0.149*	*0.182*	0.064	0.08	0.09	0.108	0.076	0.082	0.071	0.107	Emphysema				0	single	3	G*01:01,*--	UTR1,UTR4
6	0.078	0.077	NR	NR	0.058	0.061	0.052	0.059	0.051	0.056	Cystic fibrosis				3	bilateral	8	G*01:01,*--	UTR1,UTR2
7	0.06	0.065	0.056	0.058	0.051	0.051	0.061	0.067	0.061	0.067	Emphysema				2	bilateral	6	G*01:01,G*01:06	UTR2,UTR6
8	NR	NR	NR	NR	NR	NR	0.068	0.072	NR	NR	Cystic fibrosis			M1	0	bilateral	6	G*01:01,G*01:04	UTR1,UTR3
9	0.062	0.105	0.096	*0.09*	*0.262*	*0.325*	0.058	0.076	0.057	0.088	Emphysema			D0	0	bilateral	6	G*01:01,G*01:06	UTR2,UTR2
10	*0.645*	*0.349*	0.064	0.057	*0.167*	0.081	0.074	0.077	0.072	0.068	Cystic fibrosis			M3	0	bilateral	6	G*01:01,G*01:04	UTR3,UTR6
11	*0.385*	*0.394*	0.065	0.074	NR	NR	NR	NR	NR	NR	Emphysema			D15	0	bilateral	0	G*01:01,G*01:06	UTR2,UTR2
12	0.05	0.063	0.061	0.066	0.059	0.07	0.061	0.06	0.052	0.059	Cystic fibrosis	M1		D15	0	single	8	G*01:01,G*01:05N	UTR4,UTR3
13	0.102	0.103	0.064	0.062	0.088	0.084	0.07	0.057	0.066	0.072	Cystic fibrosis			M1	0	single	6	G*01:01,*--	UTR1,UTR2
14	*0.35*	0.357	NR	NR	0.057	0.06	0.061	0.059	0.09	0.068	Fibrosis			M1	0	bilateral	8	G*01:01,G*01:04	UTR3,UTR7
15	0.086	0.078	0.054	0.057	0.064	0.06	0.068	0.064	0.057	0.078	Cystic fibrosis			M1	0	bilateral	7	G*01:01,*--	UTR1,UTR2
16	0.06	0.06	NR	NR	*0.29*	*0.23*	0.06	0.06	0.06	0.07	Fibrosis				2	single	7	G*01:01,*--	UTR2,UTR6
17	0.101	0.088	0.053	0.102	0.064	0.076	0.078	0.085	0.07	0.07	Fibrosis			D15	0	bilateral	4	G*01:01,G*01:03	UTR1, UTR2
18	NR	NR	NR	NR	NR	NR	0.057	0.055	NR	NR	Cystic fibrosis			M1	0	bilateral	5	G*01:01,*--	UTR4,UTR2
19	*0.728*	*0.564*	0.134	0.104	0.112	0.094	0.081	0.073	0.08	0.074	Cystic fibrosis			D15	0	bilateral	8	G*01:03,G*01:06	UTR2,UTR5
20	0.122	0.081	*0.314*	*0.247*	0.054	0.059	0.057	0.07	0.06	0.077	Cystic fibrosis				0	single	6	G*01:01,*--	UTR1,UTR6
21	0.078	0.068	0.071	0.058	NR	NR	0.051	0.059	0.07	0.093	Fibrosis	M3	Yes	M1	0	bilateral	0	G*01:01,G*01:04	UTR4,UTR3
22	0.138	0.143	0.059	0.054	0.059	0.063	0.058	0.07	0.055	0.075	Emphysema		Yes	D15	2	bilateral	5	G*01:01,*--	UTR1,UTR2
23	*0.194*	*0.209*	0.116	*0.178*	0.118	*0.301*	0.117	*0.192*	NR	NR	Bronchial Dysplasia		Yes	D0	3	bilateral	7	G*01:01,*--	UTR4,UTR6
24	0.076	0.07	0.058	0.059	0.134	0.055	0.058	0.055	0.065	0.069	Cystic fibrosis				0	bilateral	0	G*01:01,G*01:05N	UTR1,UTR2
25	0.11	0.117	0.1	0.072	*0.194*	*0.194*	*0.418*	*0.466*	0.082	0.068	Fibrosis				0	single	6	G*01:01,*--	UTR4,UTR7
26	NR	NR	0.077	0.081	NR	NR	NR	NR	0.056	0.068	cystic Fibrosis			D15	0	bilateral	6	G*01:01,*--	UTR1,UTR2
27	0.059	0.065	0.061	0.058	0.055	0.069	0.059	0.058	0.07	0.063	Emphysema	M12	Yes	M1	0	single	7	G*01:01,*--	UTR1,UTR2
28	*0.169*	*0.208*	0.057	0.059	0.125	0.074	0.061	0.074	0.06	0.063	Cystic fibrosis	M3		M1	3	bilateral	7	G*01:01,*--	UTR2,UTR4
29	NR	NR	NR	NR	NR	NR	0.07	0.058	NR	NR	Cystic fibrosis				1	bilateral	7	G*01:01,G*01:04	UTR1,UTR3
30	0.073	0.081	NR	NR	0.078	0.096	0.058	0.067	NR	NR	Cystic fibrosis	M3		M1	2	bilateral	7	G*01:01,*--	UTR1,UTR2
31	0.083	0.077	NR	NR	0.055	0.062	0.056	0.054	0.064	0.069	Cystic fibrosis		Yes	M1	0	bilateral	8	G*01:01,*--	UTR7,UTR1
32	0.07	0.101	0.067	0.068	0.094	0.079	0.139	*0.154*	0.071	0.061	Fibrosis			D0	3	single	6	G*01:01,G*01:04	UTR1,UTR3
33	*0.61*	*0.397*	0.057	0.054	0.11	0.1	0.066	0.058	0.088	0.065	Cystic fibrosis	M3		M1	0	bilateral	0	G*01:01,G*01:06	UTR2,UTR2
34	NR	NR	NR	NR	0.06	0.066	NR	NR	0.053	0.058	Emphysema			M1	0	single	5	G*01:01,*--	UTR1,UTR7
35	0.077	0.1	NR	NR	0.076	0.084	0.063	0.076	0.08	0.131	Fibrosis		Yes	M1	0	bilateral	7	G*01:01,*--	UTR1, UTR1

## Data Availability

Data are contained within the article or Appendix A.

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
