# Peer review of "Low Prevalence of HLA-G Antibodies in Lung Transplant Patients Detected using MAIPA-Adapted Protocol"

_ijms, 2023, doi:10.3390/ijms242216479_

Round 1
Reviewer 1 Report
Comments and Suggestions for Authors
The manuscript submitted by Pascal Pedini and co-authors is devoted to analyzing HLA-G antibodies in lung transplant patients.
Several flaws do not allow the paper to be published in its current state.
1) Major flaws
1.1) The most exciting experiment in the paper uses the SPI-801 cell line. The authors should add in the Introduction the information regarding this cell line. Evidence that this cell line is relevant for such experimental design, etc. should also be added.
1.2) Sequences of cDNA and/or plasmids should be added in supplementary.
1.3) Protocol of flow cytometry, the strategy of gating, etc., should be added
1.4) Luminex and similar platforms generate lots of data, but their relevance, adequacy, and reliability are often questionable (due to many false positive and false negative results). Which control samples did the authors use to prove the results?
1.5) What is the physiological relevance of the results obtained by the authors (role of anti-HLA-G antibodies, etc.)? This should be discussed in the Results and Discussion.
1.6) What was the hypothesis the authors wanted to test?
2) Other comments
2.1) Text in lines 91-94 should be removed.
2.2) Tables 1-4 should be added as text and formatted in MDPI style, not inserted as images.
Author Response
Dr Pascal Pedini, PharmD, PhD
Department of Immunogenetic, EFS PACC
149, Boulevard Baille, 13005 Marseille, France
Phone: 33-4-91-18-95-96
FAX: 33-4-91-48-16-02
e-mail: pascal.pedini@efs.sante.fr
October 23, 2023
Revised manuscript.
Dear Chief Editor,
Please find enclosed our manuscript untitled “Low prevalence of HLA-G Antibodies in lung transplant patients detected by MAIPA-adapted protocol.», which has been revised after addressing all reviewer comments.
Reviewer 1 :
pen Review
(x) I would not like to sign my review report
( ) I would like to sign my review report
Quality of English Language
(x) I am not qualified to assess the quality of English in this paper
( ) English very difficult to understand/incomprehensible
( ) Extensive editing of English language required
( ) Moderate editing of English language required
( ) Minor editing of English language required
( ) English language fine. No issues detected
Yes Can be improved Must be improved Not applicable
Does the introduction provide sufficient background and include all relevant references? ( ) ( ) (x) ( )
Are all the cited references relevant to the research? ( ) (x) ( ) ( )
Is the research design appropriate? ( ) ( ) (x) ( )
Are the methods adequately described? ( ) ( ) (x) ( )
Are the results clearly presented? ( ) ( ) (x) ( )
Are the conclusions supported by the results? ( ) ( ) (x) ( )
Comments and Suggestions for Authors
The manuscript submitted by Pascal Pedini and co-authors is devoted to analyzing HLA-G antibodies in lung transplant patients.
Several flaws do not allow the paper to be published in its current state.
1) Major flaws
1.1) The most exciting experiment in the paper uses the SPI-801 cell line. The authors should add in the Introduction the information regarding this cell line. Evidence that this cell line is relevant for such experimental design, etc. should also be added.
The authors agree with the reviewer. We added this sentence and these references.
« This cell line was chosen because it does not express classical and non-classical HLA molecules and has already been the subject of several molecular transductions (Hayashi, T, celik AA). »
Hayashi T, Amakishi E, Matsuyama N, Yasui K, Furuta RA, Hori Y, Tanaka S, Fukumori Y, Hirayama F. Detection of anti-human platelet antibodies against integrin α2β1 using cell lines. Blood Transfus. 2014 Jan;12 Suppl 1(Suppl 1):s273-80.
Celik AA, Simper GS, Huyton T, Blasczyk R, Bade-Döding C. HLA-G mediated immune regulation is impaired by a single amino acid exchange in the alpha 2 domain. Hum Immunol. 2018 Jun;79(6):453-462.
1.2) Sequences of cDNA and/or plasmids should be added in supplementary.
The authors agree with the reviewer. We added supplementary figures including the pWPXL vector mapping (Supplementary figure 1) and the cDNA HLA-G*01:01, *01 :04, *01:06 sequences (Supplementary figure 2).
1.3) Protocol of flow cytometry, the strategy of gating, etc., should be added
The authors agree with the reviewer. We modified and added these sentences : « Flow cytometry (FCM) was performed using a Myltenyi VYB cytometer with MEM-G/9 fluorescein allophycocianin–conjugated (APC) antibody (HLA-G; Thermo Fisher #MA1-19014) or the isotype-matched control-APC Ab. HLA-G-SPI-801 cells were gated on cell populations, based upon forward and side scatter, and results were expressed as % cells expressing HLA-G above the level of fluorescence detected with the isotype control mAb. »
1.4) Luminex and similar platforms generate lots of data, but their relevance, adequacy, and reliability are often questionable (due to many false positive and false negative results). Which control samples did the authors use to prove the results?
The authors agree with the reviewer. We added these sentences and these references : « MEM-G/9 monoclonal antibody was developed against the human recombinant full length HLA-G1 protein and is today very well characterized 28. MEM-G/9 is used in HLA-G ELISA as a capture antibody 29. MEM-G/9 binds surface expressed native HLA-G1monomer and dimer but but not to denatured HLA-G1 30. MEM-G/9 only recognizes HLA-G1 isoforms associated with B2 microglobulins as shown by a mild acid treatment which negative detection. In order to know the specificity of the MEMG/9 monoclonal antibody, this reactivity was studied against peripheral cells expressing different classical HLA antigens by FCM. A slight cross-reactivities without identified HLA specificity were detected against some cells. Furthermore, MEM-G/9 antibody did not cross-react with any of the HLA-E and HLA-F expressed on SPI-8 cell lines by FCM. However, it cannot be ruled out that mem-G/9 can recognize epitopes that are present on other HLA molecules or related molecules, as has been demonstrated for HLA-E monoclonal antibodies, 3D12 and MEM-E/02, which recognizes some HLA-B and HLA-C alleles 31,32. However, MAIPA limits these cross-reactivities. Indeed, technically, the first deposition of serums on the transduced cells followed by washing limits the risk cross-contamination by circulant HLA. »
Fournel S., Huc X., Aguerre-Girr M., Solier C., Legros M., Praud-Brethenou C., Moussa M., Chaouat G., Berrebi A., Bensussan A., et al. Comparative reactivity of different HLA-G monoclonal antibodies to soluble HLA-G molecules. Tissue Antigens. 2000;55:510–518.
Zhao, L.; Teklemariam, T.; Hantash, B.M. Reassessment of HLA-G isoform specificity of MEM-G/9 and 4H84 monoclonal antibodies. Tissue Antigens 2012, 80, 231–238.
Furukawa A, Meguro M, Yamazaki R, Watanabe H, Takahashi A, Kuroki K, Maenaka K. Evaluation of the Reactivity and Receptor Competition of HLA-G Isoforms Toward Available Antibodies: Implications of Structural Characteristics of HLA-G Isoforms. Int J Mol Sci. 2019 Nov 26;20(23):5947.
Ravindranath MH, Pham T, El-Awar N, Kaneku H, Terasaki PI. Anti-HLA-E mAb 3D12 mimics MEM-E/02 in binding to HLA-B and HLA-C alleles: Web-tools validate the immunogenic epitopes of HLA-E recognized by the antibodies. Mol Immunol. 2011 Jan;48(4):423-30.
Ravindranath MH, Taniguchi M, Chen CW, Ozawa M, Kaneku H, El-Awar N, Cai J, Terasaki PI. HLA-E monoclonal antibodies recognize shared peptide sequences on classical HLA class Ia: relevance to human natural HLA antibodies. Mol Immunol. 2010 Feb;47(5):1121-31.
1.5) What is the physiological relevance of the results obtained by the authors (role of anti-HLA-G antibodies, etc.)? This should be discussed in the Results and Discussion.
The authors agree with the reviewer. We added this sentence in the Discussion :
« The absence of detection of anti-HLA-G antibodies in patients who have not had immunizing events suggests the absence of reactivity with non denaturated HLA antibodies. In addition, this result is surely more physiologically compatible with the risk of pathogenicity of these antibodies on the dysregulation of tolerance of the immune system than those obtained by Multiplex Luminex®-based flow cytometry. »
1.6) What was the hypothesis the authors wanted to test?
The authors agree with the reviewer. We added the hypothesis of this project :
« Also, these anti-HLA-G antibodies can actively participate in the occurrence or progression of CLAD in transplant patients. »
2) Other comments
2.1) Text in lines 91-94 should be removed.
The authors agree with the reviewer. We removed these lines.
2.2) Tables 1-4 should be added as text and formatted in MDPI style, not inserted as images.
The authors agree with the reviewer. The tables 1-4 have been added as text and formatted in MDPI style.
Reviewer 2 Report
Comments and Suggestions for Authors
Type of the Paper (Article
Review report on International Journal of Molecular Sciences 2611118
Title of the Article: Low prevalence of HLA-G Antibodies in lung transplant patients detected by MAIPA-adapted protocol.
Authors: Pascal Pedini, Lucas Hubert, Federico Carlini, Jean Baptiste Baudey, Audrey Tous, Francois Jordier, Agnès Basire, Claude Bagnis, Martine Reynaud-Gaubert, Benjamin Coiffard, Jacques Chiaroni, Monique Silvy, Christophe Picard.
Outline of this investigation:
The authors' objectives are
(1) to detect HLA-G antibodies in lung transplant recipients (LTRs);
(2) to correlate the occurrence of the HLA-G antibodies with the clinical complications.
For this purpose
(1) they incubated the cell lines SPI-801, obtained before or after transfecting with 3 different HLA-G isoforms expression, with sera from healthy blood donors (n = 90) and LTRs (n = 35) (before and after 24 transplantation).
(2) The presence of HLA-G on the transfected cells and their reactivities with sera antibodies were investigated using commercial monoclonal antibodies, particularly utilizing the immobilization of platelet antigen assay (MAIPA ApDIA®).
The monoclonal antibodies used include
(1) Western blot 265 was performed with 4H84 antibody (HLA-G, R&D system #MABF2169).
(2) Flow cytometry 266 was performed using a Myltenyi VYB cytometer
(A) with MEM-G/9 antibody (anti-HLA-G antibody)
(B) with 3D12 (anti-HLA-E antibody;
General comments:
This is a very interesting study and the authors have made a sincere effort to investigate their objective. Most importantly, their conclusion is striking and interesting and it deserves serious attention. The state: Only one serum from one blood donor had specific reactivity against the HLA-G transduced lines. Non-specific reactivity in many sera from LTRs was observed with transduced and wild-type cell lines, which may suggest recognition of an autoantigen expressed by the SPI-801 cell line.
Specific comments:
This particular statement of the authors deserves serious and critical examination.
One fundamental question is to what extent the authors are sure that the monoclonal antibodies (mAb 4H84 & mAb MEM-G/9) used are “Monospecific” for HLA-G and the mAb 3D12 is specific for HLA-E?
The reviewer wishes to bring to the attention of the authors the following publications:
[1] The paper was published by Late Professor Paul Ichiro Terasaki and his team of scientists on mAb 3D12. This mAb was assumed in almost all previous investigators who have used this mAb to be specific for HLA-E. In fact, it found to be the opposite of what was assumed in the following paper:
Terasaki, P.I. team paper entitled: Anti-HLA-E mAb 3D12 mimics MEM-E/02 in binding to HLA-B and HLA-C alleles: Web-tools validate the immunogenic epitopes of HLA-E recognized by the Abs. Mol. Immunol. 2011, 48, 423–430.
This reviewer insists that all of the 13 authors of the current IJMS article read this paper and realize the pitfalls in their paper while using mAb 3D12.
Serious request to authors: Do not assume, but test the mAb for monospecificity for the target antigen before extending your tentacles on the usefulness of the mAb.
[2] MEM SERIES antibodies were developed by a particular group which include mAbs for HLA-E (MEM-E02. MEM-E06, MEM-E-07, MEM-E/08) and HLA-G (MEM-G/9).
Let us see the reliability of using anti-HLA-E mAbs developed by this group. Again Terasaki group thoroughly investigated these mAbs, as they did for 3D12.
Terasaki, P.I. team paper entitled: HLA-E monoclonal antibodies recognize shared peptide sequences on classical HLA class Ia: Relevance to human natural HLA antibodies. Mol. Immunol. 2010, 47, 121–131
This reviewer insists that all of the 13 authors of the current IJMS article to read this paper and realize the pitfalls in their paper while using MEM mAbs.
Serious request to authors: Do not assume, but test the mAb for monospecificity for HLA-G before extending your tentacles on the usefulness of the mAb.
Now the reviewer asks whether MEM-G/9 is reliable for immunocapture. The authors have smartly stated “One possibility is that the MEM-G9 monoclonal antibody can cross-reactivate with epitopes carried by classical HLA”
However, the authors assume that “the selected line did not express any HLA molecule”. On what basis? Do they test antibodies against HLA-F, HLA-A, HLA-B or HLA-C or with mAb W632 or HC10?
The authors' final statement is “the results need to be confirmed by immuno-capture performed with other HLA-G monoclonal antibodies”
As stated in the introduction, one or more of the 13 authors should Multiplex Luminex®-based flow cytometry should have considered for screening mAbs directed against non-classical HLA (HLA-G, -E, and –F).
One important note to authors, Transplantation involves extensive injury-induced inflammation during which process cells get activated and HLA is expressed as a monomer not associated with B2-microglobulin. These monomers expressed on cells both in tissues and in circulating exosomes expose cryptic epitopes masked by B2M when they are intact. Such exposure induces immune recognition and production of antibodies not only against alleles-specific HLA monomers but also epitopes shared by all HLAs.
I suggest that all 13 authors should have a glance at the following papers from Terasaki Laboratory: Biomolecules 2023, 13, 1178. https://doi.org/10.3390/biom13081178; Antibodies 2022, 11, 18. doi: /10.3390/antib 11010018
Recommendations:
The reviewer is quite impressed by the honesty and integrity of the authors by concurring that their findings are ambiguous (and to some extent unreliable) and particularly and specifically they state that the results need to be confirmed by immuno-capture performed with other HLA-G monoclonal antibodies, I strongly URGE the authors to find a lab that has Multiplex Luminex Flow cytometry (such as Terasaki Research Institute, Canoga park) or send to experienced investigators like Dr. Vadim Jucaud (whom they have cited), and test MEM-G/9 for cross-reactivity with different HLAs (HLA-A, HLA-B and HLA-C and if possible with HLA-E and HLA-F) and prove that MEM-G/9 does not bind to any one of these HLAs in LABSCreen beadsets containing the admixture of B2M-associated and B2M-free HLAs. If it binds or does not bind, the authors make an outstanding contribution to the science of HLA antibodies.
Therefore the paper must be revised with Multiplex Luminex Flow cytometry data on MEM-G/9 and possibly mAb 4H84 too, for its acceptance in any journal. The manuscript richly deserves publication after this investigation on MEM-G/9.
Author Response
Dr Pascal Pedini, PharmD, PhD
Department of Immunogenetic, EFS PACC
149, Boulevard Baille, 13005 Marseille, France
Phone: 33-4-91-18-95-96
FAX: 33-4-91-48-16-02
e-mail: pascal.pedini@efs.sante.fr
October 23, 2023
Revised manuscript.
Dear Chief Editor,
Please find enclosed our manuscript untitled “Low prevalence of HLA-G Antibodies in lung transplant patients detected by MAIPA-adapted protocol.», which has been revised after addressing all reviewer comments.
Reviewer 2 :
Open Review
( ) I would not like to sign my review report
(x) I would like to sign my review report
Quality of English Language
( ) I am not qualified to assess the quality of English in this paper
( ) English very difficult to understand/incomprehensible
( ) Extensive editing of English language required
( ) Moderate editing of English language required
( ) Minor editing of English language required
(x) English language fine. No issues detected
Yes Can be improved Must be improved Not applicable
Does the introduction provide sufficient background and include all relevant references? (x) ( ) ( ) ( )
Are all the cited references relevant to the research? (x) ( ) ( ) ( )
Is the research design appropriate? ( ) ( ) (x) ( )
Are the methods adequately described? (x) ( ) ( ) ( )
Are the results clearly presented? ( ) ( ) (x) ( )
Are the conclusions supported by the results? ( ) ( ) (x) ( )
Comments and Suggestions for Authors
Type of the Paper (Article
Review report on International Journal of Molecular Sciences 2611118
Title of the Article: Low prevalence of HLA-G Antibodies in lung transplant patients detected by MAIPA-adapted protocol.
Authors: Pascal Pedini, Lucas Hubert, Federico Carlini, Jean Baptiste Baudey, Audrey Tous, Francois Jordier, Agnès Basire, Claude Bagnis, Martine Reynaud-Gaubert, Benjamin Coiffard, Jacques Chiaroni, Monique Silvy, Christophe Picard.
Outline of this investigation:
The authors' objectives are
(1) to detect HLA-G antibodies in lung transplant recipients (LTRs);
(2) to correlate the occurrence of the HLA-G antibodies with the clinical complications.
For this purpose
(1) they incubated the cell lines SPI-801, obtained before or after transfecting with 3 different HLA-G isoforms expression, with sera from healthy blood donors (n = 90) and LTRs (n = 35) (before and after 24 transplantation).
(2) The presence of HLA-G on the transfected cells and their reactivities with sera antibodies were investigated using commercial monoclonal antibodies, particularly utilizing the immobilization of platelet antigen assay (MAIPA ApDIA®).
The monoclonal antibodies used include
(1) Western blot 265 was performed with 4H84 antibody (HLA-G, R&D system #MABF2169).
(2) Flow cytometry 266 was performed using a Myltenyi VYB cytometer
(A) with MEM-G/9 antibody (anti-HLA-G antibody)
(B) with 3D12 (anti-HLA-E antibody;
General comments:
This is a very interesting study and the authors have made a sincere effort to investigate their objective. Most importantly, their conclusion is striking and interesting and it deserves serious attention. The state: Only one serum from one blood donor had specific reactivity against the HLA-G transduced lines. Non-specific reactivity in many sera from LTRs was observed with transduced and wild-type cell lines, which may suggest recognition of an autoantigen expressed by the SPI-801 cell line.
Specific comments:
This particular statement of the authors deserves serious and critical examination.
One fundamental question is to what extent the authors are sure that the monoclonal antibodies (mAb 4H84 & mAb MEM-G/9) used are “Monospecific” for HLA-G and the mAb 3D12 is specific for HLA-E?
The reviewer wishes to bring to the attention of the authors the following publications:
[1] The paper was published by Late Professor Paul Ichiro Terasaki and his team of scientists on mAb 3D12. This mAb was assumed in almost all previous investigators who have used this mAb to be specific for HLA-E. In fact, it found to be the opposite of what was assumed in the following paper:
Terasaki, P.I. team paper entitled: Anti-HLA-E mAb 3D12 mimics MEM-E/02 in binding to HLA-B and HLA-C alleles: Web-tools validate the immunogenic epitopes of HLA-E recognized by the Abs. Mol. Immunol. 2011, 48, 423–430.
This reviewer insists that all of the 13 authors of the current IJMS article read this paper and realize the pitfalls in their paper while using mAb 3D12.
Serious request to authors: Do not assume, but test the mAb for monospecificity for the target antigen before extending your tentacles on the usefulness of the mAb.
[2] MEM SERIES antibodies were developed by a particular group which include mAbs for HLA-E (MEM-E02. MEM-E06, MEM-E-07, MEM-E/08) and HLA-G (MEM-G/9).
Let us see the reliability of using anti-HLA-E mAbs developed by this group. Again Terasaki group thoroughly investigated these mAbs, as they did for 3D12.
Terasaki, P.I. team paper entitled: HLA-E monoclonal antibodies recognize shared peptide sequences on classical HLA class Ia: Relevance to human natural HLA antibodies. Mol. Immunol. 2010, 47, 121–131
This reviewer insists that all of the 13 authors of the current IJMS article to read this paper and realize the pitfalls in their paper while using MEM mAbs.
Serious request to authors: Do not assume, but test the mAb for monospecificity for HLA-G before extending your tentacles on the usefulness of the mAb.
The authors agree with the reviewer. We added these sentences and these references :
« MEM-G/9 monoclonal antibody was developed against the human recombinant full length HLA-G1 protein and is today very well characterized 28. MEM-G/9 is used in HLA-G ELISA as a capture antibody 29. MEM-G/9 binds surface expressed native HLA-G1monomer and dimer but but not to denatured HLA-G1 30. MEM-G/9 only recognizes HLA-G1 isoforms associated with B2 microglobulins as shown by a mild acid treatment which negative detection. In order to know the specificity of the MEMG/9 monoclonal antibody, this reactivity was studied against peripheral cells expressing different classical HLA antigens by FCM. A slight cross-reactivities without identified HLA specificity were detected against some cells. Furthermore, MEM-G/9 antibody did not cross-react with any of the HLA-E and HLA-F expressed on SPI-8 cell lines by FCM. However, it cannot be ruled out that mem-G/9 can recognize epitopes that are present on other HLA molecules or related molecules, as has been demonstrated for HLA-E monoclonal antibodies, 3D12 and MEM-E/02, which recognizes some HLA-B and HLA-C alleles 31,32. However, MAIPA limits these cross-reactivities. Indeed, technically, the first deposition of serums on the transduced cells followed by washing limits the risk cross-contamination by circulant HLA. »
Fournel S., Huc X., Aguerre-Girr M., Solier C., Legros M., Praud-Brethenou C., Moussa M., Chaouat G., Berrebi A., Bensussan A., et al. Comparative reactivity of different HLA-G monoclonal antibodies to soluble HLA-G molecules. Tissue Antigens. 2000;55:510–518.
Zhao, L.; Teklemariam, T.; Hantash, B.M. Reassessment of HLA-G isoform specificity of MEM-G/9 and 4H84 monoclonal antibodies. Tissue Antigens 2012, 80, 231–238.
Furukawa A, Meguro M, Yamazaki R, Watanabe H, Takahashi A, Kuroki K, Maenaka K. Evaluation of the Reactivity and Receptor Competition of HLA-G Isoforms Toward Available Antibodies: Implications of Structural Characteristics of HLA-G Isoforms. Int J Mol Sci. 2019 Nov 26;20(23):5947.
Ravindranath MH, Pham T, El-Awar N, Kaneku H, Terasaki PI. Anti-HLA-E mAb 3D12 mimics MEM-E/02 in binding to HLA-B and HLA-C alleles: Web-tools validate the immunogenic epitopes of HLA-E recognized by the antibodies. Mol Immunol. 2011 Jan;48(4):423-30.
Ravindranath MH, Taniguchi M, Chen CW, Ozawa M, Kaneku H, El-Awar N, Cai J, Terasaki PI. HLA-E monoclonal antibodies recognize shared peptide sequences on classical HLA class Ia: relevance to human natural HLA antibodies. Mol Immunol. 2010 Feb;47(5):1121-31.
Now the reviewer asks whether MEM-G/9 is reliable for immunocapture. The authors have smartly stated “One possibility is that the MEM-G9 monoclonal antibody can cross-reactivate with epitopes carried by classical HLA”
However, the authors assume that “the selected line did not express any HLA molecule”. On what basis? Do they test antibodies against HLA-F, HLA-A, HLA-B or HLA-C or with mAb W632 or HC10?
The authors agree with the reviewer. We added this sentence and these references:
“ This cell line was chosen because it does not express classical and non-classical HLA molecules and has already been the subject of several molecular transductions 22,23 “
Hayashi T, Amakishi E, Matsuyama N, Yasui K, Furuta RA, Hori Y, Tanaka S, Fukumori Y, Hirayama F. Detection of anti-human platelet antibodies against integrin α2β1 using cell lines. Blood Transfus. 2014 Jan;12 Suppl 1(Suppl 1):s273-80.
Celik AA, Simper GS, Huyton T, Blasczyk R, Bade-Döding C. HLA-G mediated immune regulation is impaired by a single amino acid exchange in the alpha 2 domain. Hum Immunol. 2018 Jun;79(6):453-462.
We tested the absence of reactivity detection by FCM using W6.32 monoclonal antibody.
The authors' final statement is “the results need to be confirmed by immuno-capture performed with other HLA-G monoclonal antibodies”
As stated in the introduction, one or more of the 13 authors should Multiplex Luminex®-based flow cytometry should have considered for screening mAbs directed against non-classical HLA (HLA-G, -E, and –F).
One important note to authors, Transplantation involves extensive injury-induced inflammation during which process cells get activated and HLA is expressed as a monomer not associated with B2-microglobulin. These monomers expressed on cells both in tissues and in circulating exosomes expose cryptic epitopes masked by B2M when they are intact. Such exposure induces immune recognition and production of antibodies not only against alleles-specific HLA monomers but also epitopes shared by all HLAs.
I suggest that all 13 authors should have a glance at the following papers from Terasaki Laboratory: Biomolecules 2023, 13, 1178. https://doi.org/10.3390/biom13081178; Antibodies 2022, 11, 18. doi: /10.3390/antib 11010018
The authors agree with the reviewer. We added these sentences and this reference.
« Finally, transplantation involves extensive injury-induced inflammation during which process cells get activated and HLA is expressed as a monomer not associated with B2-microglobulin. These monomers expressed on cells both in tissues and in circulating exosomes expose cryptic epitopes masked by B2M when they are intact. Such exposure induces immune recognition and production of antibodies not only against alleles-specific HLA monomers but also epitopes shared by all HLAs 33. Our adapted MAIPA is not able of detecting these types of HLA antibodies, MEM-G9 binding only to non denatured HLA-G monomer or dimer. At opposite, multiplex Luminex®-based flow cytometry can detect these antibodies, the antigens fixed on the beads can be partly denatured and have a different structure. »
Ravindranath MH, Ravindranath NM, Selvan SR, Hilali FE, Amato-Menker CJ, Filippone EJ. Cell Surface B2m-Free Human Leukocyte Antigen (HLA) Monomers and Dimers: Are They Neo-HLA Class and Proto-HLA? Biomolecules. 2023 Jul 28;13(8):1178.
Recommendations:
The reviewer is quite impressed by the honesty and integrity of the authors by concurring that their findings are ambiguous (and to some extent unreliable) and particularly and specifically they state that the results need to be confirmed by immuno-capture performed with other HLA-G monoclonal antibodies, I strongly URGE the authors to find a lab that has Multiplex Luminex Flow cytometry (such as Terasaki Research Institute, Canoga park) or send to experienced investigators like Dr. Vadim Jucaud (whom they have cited), and test MEM-G/9 for cross-reactivity with different HLAs (HLA-A, HLA-B and HLA-C and if possible with HLA-E and HLA-F) and prove that MEM-G/9 does not bind to any one of these HLAs in LABSCreen beadsets containing the admixture of B2M-associated and B2M-free HLAs. If it binds or does not bind, the authors make an outstanding contribution to the science of HLA antibodies.
Therefore the paper must be revised with Multiplex Luminex Flow cytometry data on MEM-G/9 and possibly mAb 4H84 too, for its acceptance in any journal. The manuscript richly deserves publication after this investigation on MEM-G/9.
The authors agree the reviewer. Indeed, it would have been interesting to compare the results obtained by the adapted MAIPA technique with those of Multiplex Luminex Flow cytometry. One of the authors was the reviewer of Vadim Jucaud's thesis. This study was set up after discussion with Vadim, during the numerous exchanges which took place during his thesis. Several requests by email for serums in order to compare the results obtained by the luminex from Terasaki Laboratory and ours as well as other clones were requested as part of a collaboration but without success. However, the authors want to clarify that beads expressing HLA-G are not available and that the MAIPA technique used in this study precisely limits the "non-specificity" defects brought by the luminex technique. It also seemed more essential to us to analyze the specificity of MEM-G/9 in a cellular context rather than in an in vitro context where quality controls are difficult to implement.
We added these sentences and these references:
«MEM-G/9 monoclonal antibody was developed against the human recombinant full length HLA-G1 protein and is today very well characterized 28. MEM-G/9 is used in HLA-G ELISA as a capture antibody 29. MEM-G/9 binds surface expressed native HLA-G1monomer and dimer but but not to denatured HLA-G1 30. MEM-G/9 only recognizes HLA-G1 isoforms associated with B2 microglobulins as shown by a mild acid treatment which negative detection. In order to know the specificity of the MEMG/9 monoclonal antibody, this reactivity was studied against peripheral cells expressing different classical HLA antigens by FCM. A slight cross-reactivities without identified HLA specificity were detected against some cells. Furthermore, MEM-G/9 antibody did not cross-react with any of the HLA-E and HLA-F expressed on SPI-8 cell lines by FCM. However, it cannot be ruled out that mem-G/9 can recognize epitopes that are present on other HLA molecules or related molecules, as has been demonstrated for HLA-E monoclonal antibodies, 3D12 and MEM-E/02, which recognizes some HLA-B and HLA-C alleles 31,32. However, MAIPA limits these cross-reactivities. Indeed, technically, the first deposition of serums on the transduced cells followed by washing limits the risk cross-contamination by circulant HLA.»
Fournel S., Huc X., Aguerre-Girr M., Solier C., Legros M., Praud-Brethenou C., Moussa M., Chaouat G., Berrebi A., Bensussan A., et al. Comparative reactivity of different HLA-G monoclonal antibodies to soluble HLA-G molecules. Tissue Antigens. 2000;55:510–518.
Zhao, L.; Teklemariam, T.; Hantash, B.M. Reassessment of HLA-G isoform specificity of MEM-G/9 and 4H84 monoclonal antibodies. Tissue Antigens 2012, 80, 231–238.
Furukawa A, Meguro M, Yamazaki R, Watanabe H, Takahashi A, Kuroki K, Maenaka K. Evaluation of the Reactivity and Receptor Competition of HLA-G Isoforms Toward Available Antibodies: Implications of Structural Characteristics of HLA-G Isoforms. Int J Mol Sci. 2019 Nov 26;20(23):5947.
Ravindranath MH, Pham T, El-Awar N, Kaneku H, Terasaki PI. Anti-HLA-E mAb 3D12 mimics MEM-E/02 in binding to HLA-B and HLA-C alleles: Web-tools validate the immunogenic epitopes of HLA-E recognized by the antibodies. Mol Immunol. 2011 Jan;48(4):423-30.
Ravindranath MH, Taniguchi M, Chen CW, Ozawa M, Kaneku H, El-Awar N, Cai J, Terasaki PI. HLA-E monoclonal antibodies recognize shared peptide sequences on classical HLA class Ia: relevance to human natural HLA antibodies. Mol Immunol. 2010 Feb;47(5):1121-31.
We performed many analyzes of the specificity of the MEM-G9 clone by FCM on several donor cells as well as on SPi-8 cells (data transmitted).
Reviewer 3 Report
Comments and Suggestions for Authors
Dear authors,
First of all, I would like to commend you for undertaking a study that explores the complex role of HLA-G antibodies. The topic is undoubtedly extremely important and efforts to clarify it further are noteworthy. In this article, I present my review, highlighting both the strengths and areas where there may be room for greater clarity and depth.
Summary:
Part of the article discusses an experimental study on the detection of HLA-G
Strengths:
- The study broadly assessed the presence of HLA-G antibodies in different people, providing a holistic view of the topic.
- The use of the MAIPA ApDIA® kit and other tools such as the Luminex method demonstrates the detailed methodological approach used.
- The authors compare their results with previous findings, especially those of Jucaud et al. studies, providing an in-depth analysis of existing knowledge on a given topic.
- The paper acknowledges the potential limitations and limitations of their methods, providing clarity.
Areas requiring improvement:
- Despite its technical depth, the study's results contradict some of the existing literature. The reasons for these discrepancies need to be further investigated.
- The potential for cross-reactivity in antibody detection has been mentioned. It would be beneficial to investigate this issue further as it could impact the study's conclusions.
- More details about the selected cohort, such as inclusion and exclusion criteria, would provide more context and depth to the reader.
- It has been mentioned that the MAIPA test may be less sensitive than the Luminex method. The implications of this difference in sensitivity need to be explored in more detail.
Application:
This section of the article offers a comprehensive examination of the role of HLA-G antibodies in LTx outcomes. Although the findings suggest a minimal role for these antibodies in post-transplant complications, further studies are warranted to confirm these results. Given its technical depth and consideration of prior literature, this segment makes a significant contribution to knowledge on the topic.
Author Response
Dr Pascal Pedini, PharmD, PhD
Department of Immunogenetic, EFS PACC
149, Boulevard Baille, 13005 Marseille, France
Phone: 33-4-91-18-95-96
FAX: 33-4-91-48-16-02
e-mail: pascal.pedini@efs.sante.fr
October 23, 2023
Revised manuscript.
Dear Chief Editor,
Please find enclosed our manuscript untitled “Low prevalence of HLA-G Antibodies in lung transplant patients detected by MAIPA-adapted protocol.», which has been revised after addressing all reviewer comments.
Reviewer 3 :
Open Review
(x) I would not like to sign my review report
( ) I would like to sign my review report
Quality of English Language
(x) I am not qualified to assess the quality of English in this paper
( ) English very difficult to understand/incomprehensible
( ) Extensive editing of English language required
( ) Moderate editing of English language required
( ) Minor editing of English language required
( ) English language fine. No issues detected
Yes Can be improved Must be improved Not applicable
Does the introduction provide sufficient background and include all relevant references? ( ) ( ) (x) ( )
Are all the cited references relevant to the research? ( ) ( ) (x) ( )
Is the research design appropriate? ( ) ( ) (x) ( )
Are the methods adequately described? ( ) ( ) (x) ( )
Are the results clearly presented? ( ) ( ) (x) ( )
Are the conclusions supported by the results? ( ) ( ) (x) ( )
Comments and Suggestions for Authors
Dear authors,
First of all, I would like to commend you for undertaking a study that explores the complex role of HLA-G antibodies. The topic is undoubtedly extremely important and efforts to clarify it further are noteworthy. In this article, I present my review, highlighting both the strengths and areas where there may be room for greater clarity and depth.
Summary:
Part of the article discusses an experimental study on the detection of HLA-G
Strengths:
- The study broadly assessed the presence of HLA-G antibodies in different people, providing a holistic view of the topic.
- The use of the MAIPA ApDIA® kit and other tools such as the Luminex method demonstrates the detailed methodological approach used.
- The authors compare their results with previous findings, especially those of Jucaud et al. studies, providing an in-depth analysis of existing knowledge on a given topic.
- The paper acknowledges the potential limitations and limitations of their methods, providing clarity.
Areas requiring improvement:
- Despite its technical depth, the study's results contradict some of the existing literature. The reasons for these discrepancies need to be further investigated.
The authors agree with the reviewer. We added these sentences and these references :
« Finally, transplantation involves extensive injury-induced inflammation during which process cells get activated and HLA is expressed as a monomer not associated with B2-microglobulin. These monomers expressed on cells both in tissues and in circulating exosomes expose cryptic epitopes masked by B2M when they are intact. Such exposure induces immune recognition and production of antibodies not only against alleles-specific HLA monomers but also epitopes shared by all HLAs 33. Our adapted MAIPA is not able of detecting these types of HLA antibodies, MEM-G9 binding only to non denatured HLA-G monomer or dimer. At opposite, multiplex Luminex®-based flow cytometry can detect these antibodies, the antigens fixed on the beads can be partly denatured and have a different structure. »
Ravindranath MH, Ravindranath NM, Selvan SR, Hilali FE, Amato-Menker CJ, Filippone EJ. Cell Surface B2m-Free Human Leukocyte Antigen (HLA) Monomers and Dimers: Are They Neo-HLA Class and Proto-HLA? Biomolecules. 2023 Jul 28;13(8):1178.
- The potential for cross-reactivity in antibody detection has been mentioned. It would be beneficial to investigate this issue further as it could impact the study's conclusions.
The authors agree with the reviewer. We added these sentence and these references :
“MEM-G/9 monoclonal antibody was developed against the human recombinant full length HLA-G1 protein and is today very well characterized 28. MEM-G/9 is used in HLA-G ELISA as a capture antibody 29. MEM-G/9 binds surface expressed native HLA-G1monomer and dimer but but not to denatured HLA-G1 30. MEM-G/9 only recognizes HLA-G1 isoforms associated with B2 microglobulins as shown by a mild acid treatment which negative detection. In order to know the specificity of the MEMG/9 monoclonal antibody, this reactivity was studied against peripheral cells expressing different classical HLA antigens by FCM. A slight cross-reactivities without identified HLA specificity were detected against some cells. Furthermore, MEM-G/9 antibody did not cross-react with any of the HLA-E and HLA-F expressed on SPI-8 cell lines by FCM. However, it cannot be ruled out that mem-G/9 can recognize epitopes that are present on other HLA molecules or related molecules, as has been demonstrated for HLA-E monoclonal antibodies, 3D12 and MEM-E/02, which recognizes some HLA-B and HLA-C alleles 31,32. However, MAIPA limits these cross-reactivities. Indeed, technically, the first deposition of serums on the transduced cells followed by washing limits the risk cross-contamination by circulant HLA.”
Fournel S., Huc X., Aguerre-Girr M., Solier C., Legros M., Praud-Brethenou C., Moussa M., Chaouat G., Berrebi A., Bensussan A., et al. Comparative reactivity of different HLA-G monoclonal antibodies to soluble HLA-G molecules. Tissue Antigens. 2000;55:510–518.
Zhao, L.; Teklemariam, T.; Hantash, B.M. Reassessment of HLA-G isoform specificity of MEM-G/9 and 4H84 monoclonal antibodies. Tissue Antigens 2012, 80, 231–238.
Furukawa A, Meguro M, Yamazaki R, Watanabe H, Takahashi A, Kuroki K, Maenaka K. Evaluation of the Reactivity and Receptor Competition of HLA-G Isoforms Toward Available Antibodies: Implications of Structural Characteristics of HLA-G Isoforms. Int J Mol Sci. 2019 Nov 26;20(23):5947.
Ravindranath MH, Pham T, El-Awar N, Kaneku H, Terasaki PI. Anti-HLA-E mAb 3D12 mimics MEM-E/02 in binding to HLA-B and HLA-C alleles: Web-tools validate the immunogenic epitopes of HLA-E recognized by the antibodies. Mol Immunol. 2011 Jan;48(4):423-30.
Ravindranath MH, Taniguchi M, Chen CW, Ozawa M, Kaneku H, El-Awar N, Cai J, Terasaki PI. HLA-E monoclonal antibodies recognize shared peptide sequences on classical HLA class Ia: relevance to human natural HLA antibodies. Mol Immunol. 2010 Feb;47(5):1121-31.
- More details about the selected cohort, such as inclusion and exclusion criteria, would provide more context and depth to the reader.
The authors agree with the reviewer. We added these sentences :
« Patients were longitudinally followed every 6 months after LT. Inclusion criteria included at least 3 years of follow-up unless the diagnosis of CLAD was made before 3 years. Patients who died within the first 3 months posttransplantation or of a non–CLAD-related cause were excluded. »
- It has been mentioned that the MAIPA test may be less sensitive than the Luminex method. The implications of this difference in sensitivity need to be explored in more detail.
The authors agree with the reviewer. We modified this sentence and added this reference :
« MAIPA test may be less sensitive than the Luminex method 24. However, classical HLA class I alloantibodies detection from platelet lysis by the MAIPA technique (ApDIA®) is as sensitive as the Luminex technique 24. HLA-G antibody calibrated standard would allow sensitivity comparison. »
Dutra VF, Costa TH, Santos LD, Sirianni MFM, Aravechia MG, Kutner JM, Bub CB. Platelet antibodies identification: comparison between two laboratory tests. Hematol Transfus Cell Ther. 2022 Jul-Sep;44(3):365-368.
Application:
This section of the article offers a comprehensive examination of the role of HLA-G antibodies in LTx outcomes. Although the findings suggest a minimal role for these antibodies in post-transplant complications, further studies are warranted to confirm these results. Given its technical depth and consideration of prior literature, this segment makes a significant contribution to knowledge on the topic.
Round 2
Reviewer 1 Report
Comments and Suggestions for Authors
The authors responded on the reviewer's comments. The manuscript might be published after formatting Tables in MDPI style
Author Response
Dr Pascal Pedini, PharmD, PhD
Department of Immunogenetic, EFS PACC
149, Boulevard Baille, 13005 Marseille, France
Phone: 33-4-91-18-95-96
FAX: 33-4-91-48-16-02
e-mail: pascal.pedini@efs.sante.fr
November 12, 2023
Revised manuscript.
Dear Chief Editor,
Please find enclosed our manuscript untitled “Low prevalence of HLA-G Antibodies in lung transplant patients detected by MAIPA-adapted protocol.», which has been revised after addressing all reviewer comments.
Reviewer 1 :
Open Review
(x) I would not like to sign my review report
( ) I would like to sign my review report
Quality of English Language
(x) I am not qualified to assess the quality of English in this paper
( ) English very difficult to understand/incomprehensible
( ) Extensive editing of English language required
( ) Moderate editing of English language required
( ) Minor editing of English language required
( ) English language fine. No issues detected
|
Yes |
Can be improved |
Must be improved |
Not applicable |
|
|
Does the introduction provide sufficient background and include all relevant references? |
( ) |
(x) |
( ) |
( ) |
|
Are all the cited references relevant to the research? |
( ) |
(x) |
( ) |
( ) |
|
Is the research design appropriate? |
( ) |
(x) |
( ) |
( ) |
|
Are the methods adequately described? |
( ) |
(x) |
( ) |
( ) |
|
Are the results clearly presented? |
( ) |
(x) |
( ) |
( ) |
|
Are the conclusions supported by the results? |
( ) |
(x) |
( ) |
( ) |
Comments and Suggestions for Authors
The authors responded on the reviewer's comments. The manuscript might be published after formatting Tables in MDPI style
We thank the reviewer. Indeed, we have corrected the tables to match the journal's requirements
Reviewer 2 Report
Comments and Suggestions for Authors
Author Response
Dr Pascal Pedini, PharmD, PhD
Department of Immunogenetic, EFS PACC
149, Boulevard Baille, 13005 Marseille, France
Phone: 33-4-91-18-95-96
FAX: 33-4-91-48-16-02
e-mail: pascal.pedini@efs.sante.fr
November 12, 2023
Revised manuscript.
Dear Chief Editor,
Please find enclosed our manuscript untitled “Low prevalence of HLA-G Antibodies in lung transplant patients detected by MAIPA-adapted protocol.», which has been revised after addressing all reviewer comments.
Reviewer 2 :
Open Review
( ) I would not like to sign my review report
(x) I would like to sign my review report
Quality of English Language
( ) I am not qualified to assess the quality of English in this paper
( ) English very difficult to understand/incomprehensible
( ) Extensive editing of English language required
( ) Moderate editing of English language required
( ) Minor editing of English language required
(x) English language fine. No issues detected
|
Yes |
Can be improved |
Must be improved |
Not applicable |
|
|
Does the introduction provide sufficient background and include all relevant references? |
(x) |
( ) |
( ) |
( ) |
|
Are all the cited references relevant to the research? |
( ) |
(x) |
( ) |
( ) |
|
Is the research design appropriate? |
( ) |
( ) |
(x) |
( ) |
|
Are the methods adequately described? |
( ) |
( ) |
(x) |
( ) |
|
Are the results clearly presented? |
( ) |
( ) |
(x) |
( ) |
|
Are the conclusions supported by the results? |
( ) |
( ) |
(x) |
( ) |
Comments and Suggestions for Authors
- MEM-G/9 monoclonal antibody was developed against the human recombinant full length HLA-G1 protein and is today very well characterized 28.
It is therefore clear that the mAb MEM-G9 was not developed against β2m-associated HLA-G Heavy chain. Another aspect that is clear from their statement is that MEM-G/9 was developed from deglycosylated HLA-G Heavy chain, since they used recombinant full length protein.
The authors partially agree with the reviewer. We have this sentence:
MEM-G/9 monoclonal antibody was developed against the human recombinant full length HLA-G1 protein refolded with beta2-microglobulin and peptide and is today very well characterized.
- MEM-G/9 binds surface expressed HLA-G1 monomer and dimer30 Where? On cell surface or on those coated on solid matrix? Are authors fully aware what is HLA-G1 monomer and dimer? Monomer is 2m-free HLA-G heavy chain (HC). What is HLA0G1 dimer? It is paired monomers. The pairing may bring about binding between cysteine residues at different positions. Pairing is also possible between tyrosine leading to dityrosine linkage. As 2m in intact HLA-G masks several epitopes on the HC, the dimerization also leads to masking of several epitopes on the heavy chain or monomers. (Ref the following)
Ravindranath, M.H.; Ravindranath, N.M.; Selvan, S.R.; Hilali, E.F.; Amato-Menker, C.J.; Filippone,
E.J. Cell Surface B2m-Free Human Leukocyte Antigen Monomers and Dimers: Are They Neo-HLA
Class and Proto-HLA? Biomolecules 2023, 13, x. https://doi.org/10.3390/xxxxx
If the mAb–G/9 binds to both monomers and dimers, it may bind to exposed epitopes on the dimers.
The fundamental question still remains is whether mAb-G/9 recognizing epitopes are HLA-G specific
or not?
The authors agree with the reviewer. We added this sentence :
MEM-G/9 binds cell surface expressed HLA-G1 monomer and dimer, HLA-G5 isoform and also HLA-G3 isoform, suggesting that its epitope is present on the α1 domain of HLA-G [27]
- MEM-G/9 only recognizes HLA-G1 isoforms associated with 2 microglobulins as shown by a mild
acid treatment which negative detection. Where? On cell surface or on those coated on solid matrix?
That implies that the mAb MEM-G/9 can bind to b2m-associated HC or monomeric HC, or even to
dimeric HCs. If MEM-G/9 is tested on HLA-G coated beads coated with (i) b2m-associated HC; (ii)
monomeric HC and (iii) dimerized HCs, it can be confirmed that the epitope of MEM-G/9 may be an
unmasked epitope.
Negativity to acid treatment does not mean intact HLA-G becomes monomerized but the folds are
altered to mask the epitope.
The authors agree with the reviewer. We added this sentence.
MEM-G/9 only recognizes HLA-G1 isoforms associated with 2 microglobulins as shown by a mild acid treatment, which negative detection on cell surface. However, these characterizations cannot rule out that MEM-G/9 may detect an epitope that may be masked by different isoforms.
- In order to know the specificity of the MEMG/9 monoclonal antibody, this reactivity was studied
against peripheral cells expressing different classical HLA antigens by FCM. A slight crossreactivities without identified HLA specificity were detected against some cells.
What do authors mean by “slight cross reactivity”? How many different HLA class I molecules can
be found on a particular cell line? At maximum, 2 HLA-A, 2 HLA-B, 2 HLA-C, 2 HLA-E and 2 HLAF, maximum 20 HLA alleles or minimum 10 HLA alleles. The expression of these alleles with or
without 2m- depends on whether the cells are in activated state or not. If they were not activated
only slight cross reactivity can be observed.
If the author test MEM-G/9 on ONE LAMBDA LABScreen Bead set coated 90 HLA-I alleles (not
including HLA-E or F or G), they would have found out whether MEM-G/9 specific or not.
The authors agree with the reviewer. We tested MEM-G/9 on ONE LAMBDA LABScreen Bead and Immucor Lifecode ® beads following the method below :
Luminex Protocol Adapted for DxFlex
Monoclonal antibodies MEM-G/9 conjugated to PE (ThermoFisher, ref MA1-19643), W632 anti-HU HLA-ABC conjugated to PE (eBioscience), and control isotype mouse IgG conjugated to PE (BioLegend) were diluted to the recommended concentration of use (5 micrograms/mL) in negative serum. These samples were processed following the initial steps of the One Lambda (OL) Single Antigen protocol and Immucor Single Antigen protocol. In summary, 5 microL of class I identification beads were incubated with 20 microL of the sample for 30 minutes with agitation. Subsequently, five washes were performed using OL Wash Buffer or Immucor Wash Buffer. The beads were then resuspended in PBS, and the suspension was analyzed using flow cytometry on the DxFlex.
While clone W6.32 binds to all HLA class I beads, no binding is observed for MEM-G/9 with an intensity similar to that of the isotype for all beads.
We added these results in supplementary figure 1.
We have removed this sentence and replaced it with these sentences :
Moreover, no reactivity of MEM-G/9 was demonstrated on the HLA class I beads fom ONE LAMBDA LABScreen and Immucor Lifecode® (Supplementary Figure).
These data suggest that, unlike HLA-E monoclonal antibodies, 3D12 and MEM-E/02, MEM-G/9 does not cross-react with classical HLA class I molecules most frequent in the European population and that its epitope may be specific to the HLA-G molecule.
- However, it cannot be ruled out that mem-G/9 can recognize epitopes that are present on other
HLA molecules or related molecules, as has been demonstrated for HLA-E monoclonal antibodies,
3D12 and MEM-E/02, which recognizes some HLA-B and HLA-C alleles 31,32
The reviewer is happy with this confession of the authors. They can substantiate it by testing MEMG/9 on ONE LAMBDA LABScreen Bead set coated 90 HLA-I alleles (not including HLA-E or F or G),
The authors agree with the reviewer. We have removed this sentence and replaced it with these sentences :
Moreover, nNo reactivity of MEM-G/9 was demonstrated on the HLA class I beads fom ONE LAMBDA LABScreen and Immucor Lifecode ® (Supplementary Figure 1). These data suggest that, unlike HLA-E monoclonal antibodies, 3D12 and MEM-E/02, MEM-G/9 does not cross-react with classical HLA class I molecules most frequent in the European population and that its epitope may be specific to the HLA-G molecule.
- However, MAIPA limits these cross-reactivities. Indeed, technically, the first deposition of serums
on the transduced cells followed by washing limits the risk cross-contamination by circulant HLA.
On what basis the authors “conclude” that MAIPA limits these cross-reactivities. There is no evidence
for this assumption. Particularly based on the statement made on comment # 5 (in red) (it cannot be
ruled out that mem-G/9 can recognize epitopes that are present on other HLA molecules), the above
“conclusion” is apparently false or erroneous.
The authors did not agree with the reviewer. Indeed, only HLA-G molecules are expressed by cell lines. The serum could provide soluble HLA class I molecules. However, the deposition of serum on the cells before washing limits the supply of these soluble HLA molecules which could lead to cross-reactivity even if very limited by the absence of cross-reactivity demonstrated on the beads and the cells tested. For these rasons, we do not modify this sentence.
Lastly, the authors made an introductory statement one of the author was the reviewer of “Jucaud’
s” thesis. All publications of Jucaud reveal that the author has used ONE LAMBDA LABScreen Bead
set coated 90 HLA-I alleles. Therefore, the authors should contact Dr. Jucaud or any other scientist
using Lupines LABScreen bead sets coated with HLA-I alleles to test commercially available MEMG/9 on the bead sets and provide the present authors the results.
This is the only honest alternate pathway to find out whether MEM-G9 cross reacts with other HLA
I molecules such as HLA-A and/or HLA-B and/or HLA-C. It they should the detailed and honest data
in their results I have no hesitation to accept the paper with minor revision. They can even assure coauthorship to those scientist who help in this investigation
The authors agree with the reviewer. We have added it with these sentences:
Moreover, no reactivity of MEM-G/9 was demonstrated on the HLA class I beads fom ONE LAMBDA LABScreen and Immucor Lifecode ® (Supplementary Figure). These data suggest that, unlike HLA-E monoclonal antibodies, 3D12 and MEM-E/02,, MEM-G/9 does not cross-react with classical HLA class I molecules most frequent in the European population and that its epitope may be specific to the HLA-G molecule.
FINAL RECOMMENDATION.
Therefore the revised paper must be re-revised with Multiplex Luminex Flow cytometry data on
MEM-G/9 and possibly mAb 4H84 too, for its acceptance in any journal. The manuscript richly
deserves publication after this investigation on MEM-G/9.
Allowing this paper without Multiplex Luminex Flow cytometry data on MEM-G/9 is like adding
another junk publication into the current literature.
The authors thank the reviewer for all the help he provided for a better understanding of the article as well as for ensuring the probity of the results. The authors would like to remind the reviewer that the laboratory, which produced this article, is specialized in the study and detection of non-classical HLAs, especially in organ transplantations. Many publications document this knowledge as well as the authors' participation in the HLA-G and EFI congresses.
Reviewer 3 Report
Comments and Suggestions for Authors
Dear Authors,
Thank you for making corrections to your manuscript. The work already looks much better, but it would be good to make a few editorial corrections related to the formatting of the tables because they deviate from the guidelines of the IJMS journal. Additionally, when it comes to citation numbers, if we have two citation numbers next to each other, e.g. [9], [10], we can write them in one bracket [9,10] - lines 52, 56, etc.
After making the above changes, the article can be published, for my part I see no contraindications.
Author Response
Dr Pascal Pedini, PharmD, PhD
Department of Immunogenetic, EFS PACC
149, Boulevard Baille, 13005 Marseille, France
Phone: 33-4-91-18-95-96
FAX: 33-4-91-48-16-02
e-mail: pascal.pedini@efs.sante.fr
November 12 , 2023
Revised manuscript.
Dear Chief Editor,
Please find enclosed our manuscript untitled “Low prevalence of HLA-G Antibodies in lung transplant patients detected by MAIPA-adapted protocol.», which has been revised after addressing all reviewer comments.
Open Review
(x) I would not like to sign my review report
( ) I would like to sign my review report
Quality of English Language
(x) I am not qualified to assess the quality of English in this paper
( ) English very difficult to understand/incomprehensible
( ) Extensive editing of English language required
( ) Moderate editing of English language required
( ) Minor editing of English language required
( ) English language fine. No issues detected
|
Yes |
Can be improved |
Must be improved |
Not applicable |
|
|
Does the introduction provide sufficient background and include all relevant references? |
(x) |
( ) |
( ) |
( ) |
|
Are all the cited references relevant to the research? |
(x) |
( ) |
( ) |
( ) |
|
Is the research design appropriate? |
( ) |
(x) |
( ) |
( ) |
|
Are the methods adequately described? |
(x) |
( ) |
( ) |
( ) |
|
Are the results clearly presented? |
( ) |
(x) |
( ) |
( ) |
|
Are the conclusions supported by the results? |
(x) |
( ) |
( ) |
( ) |
Comments and Suggestions for Authors
Dear Authors,
Thank you for making corrections to your manuscript. The work already looks much better, but it would be good to make a few editorial corrections related to the formatting of the tables because they deviate from the guidelines of the IJMS journal. Additionally, when it comes to citation numbers, if we have two citation numbers next to each other, e.g. [9], [10], we can write them in one bracket [9,10] - lines 52, 56, etc.
After making the above changes, the article can be published, for my part I see no contraindications.
We thank the reviewer. Indeed, we have corrected the tables to match the journal's requirements and we have revised the numbering of citations when several references are side by side, such as [8, 9] , [11, 12], [8, 13, 14], [15, 16] , [15, 17, 18], [19, 20], [29, 30], [31, 32].
Round 3
Reviewer 2 Report
Comments and Suggestions for Authors
The manuscript has been well revised and is suitable for publication in its present format